# Dynamics of the fcc-to-bcc phase transition in single-crystalline PdCu alloy nanoparticles

Yingying Jiang[1,2], Martial Duchamp[3], Shi Jun Ang [4], Hongwei Yan[1,2], Teck Leong Tan [4] & Utkur Mirsaidov [1,2,5,6] ✉

Two most common crystal structures in metals and metal alloys are body-centered cubic (bcc) and face-centered cubic (fcc) structures. The phase transitions between these structures play an important role in the production of durable and functional metal alloys. Despite their technological significance, the details of such phase transitions are largely unknown because of the challenges associated with probing these processes. Here, we describe the nanoscopic details of an fcc-to-bcc phase transition in PdCu alloy nanoparticles (NPs) using in situ heating transmission electron microscopy. Our observations reveal that the bcc phase always nucleates from the edge of the fcc NP, and then propagates across the NP by forming a distinct few-atoms-wide coherent bcc–fcc interface. Notably, this interface acts as an intermediate precursor phase for the nucleation of a bcc phase. These insights into the fcc-to-bcc phase transition are important for understanding solid − solid phase transitions in general and can help to tailor the functional properties of metals and their alloys.

A solid−solid (s−s) phase transition is the transformation of a solid from one crystal structure into another[1], and such a transition is common to many minerals[2], metals[3], alloys[3], and ceramics[4−6]. Because the crystal structure of a solid determines its mechanical strength[7,8], optical property[9], and electrical[10] and thermal[11] conductivities, s−s transitions are among important processes in materials technology[12]. In metals and alloys, body-centered cubic (bcc) and face-centered cubic (fcc) structures are the two most common crystal structures. The fcc-to-bcc phase transition and vice versa are often used in the production of durable steels[12,13], shape-memory alloys[14], high entropy alloys[15], and catalytic materials[16]. However, despite their technological significance, the details of these phase transition are largely unknown because exploring the nanoscopic intermediate stages of the transitions is both experimentally and theoretically challenging[17,18].

Common experimental techniques to study phase transitions are X-ray diffraction (XRD)[19,20], electron backscatter diffraction (EBSD)[21,22], and conventional transmission electron microscopy (TEM)[15,23]. These experimental approaches have several limitations. For example, XRD can only assess the averaged crystallinity of materials and is blind to subtle intra-particle transformations. More importantly, in the case of martensitic transformations[24], which is a critical process step in steel production (e.g., displacive fcc-to-bcc transformations), the phase transition occurs at ultrahigh speeds[12,25,26]. Thus, it is almost impossible to capture the dynamics of the transition with conventional EBSD and TEM approaches, which lack the necessary temporal resolution[15,21]. Furthermore, molecular dynamics simulations[27−31], often used for the theoretical description of these transitions, provide very limited insights because the simulations employ oversimplified interatomic potentials[18].

The crucial question for understanding fcc-to-bcc phase transitions is how one phase nucleates and grows within the other phase. To address this question, we examined how a single-crystalline PdCu alloy NP transforms from an fcc into a bcc NP using in situ heating TEM

---

[1]Department of Physics, National University of Singapore, Singapore 117551, Singapore. [2]Centre for BioImaging Sciences, Department of Biological Sciences, National University of Singapore, Singapore 117557, Singapore. [3]School of Materials Science and Engineering, Nanyang Technological University, Singapore 639798, Singapore. [4]Institute of High Performance Computing, Agency for Science, Technology and Research, Singapore 138632, Singapore. [5]Centre for Advanced 2D Materials and Graphene Research Centre, National University of Singapore, Singapore 117546, Singapore. [6]Department of Materials Science and Engineering, National University of Singapore, Singapore 117575, Singapore. ✉e-mail: mirsaidov@nus.edu.sg

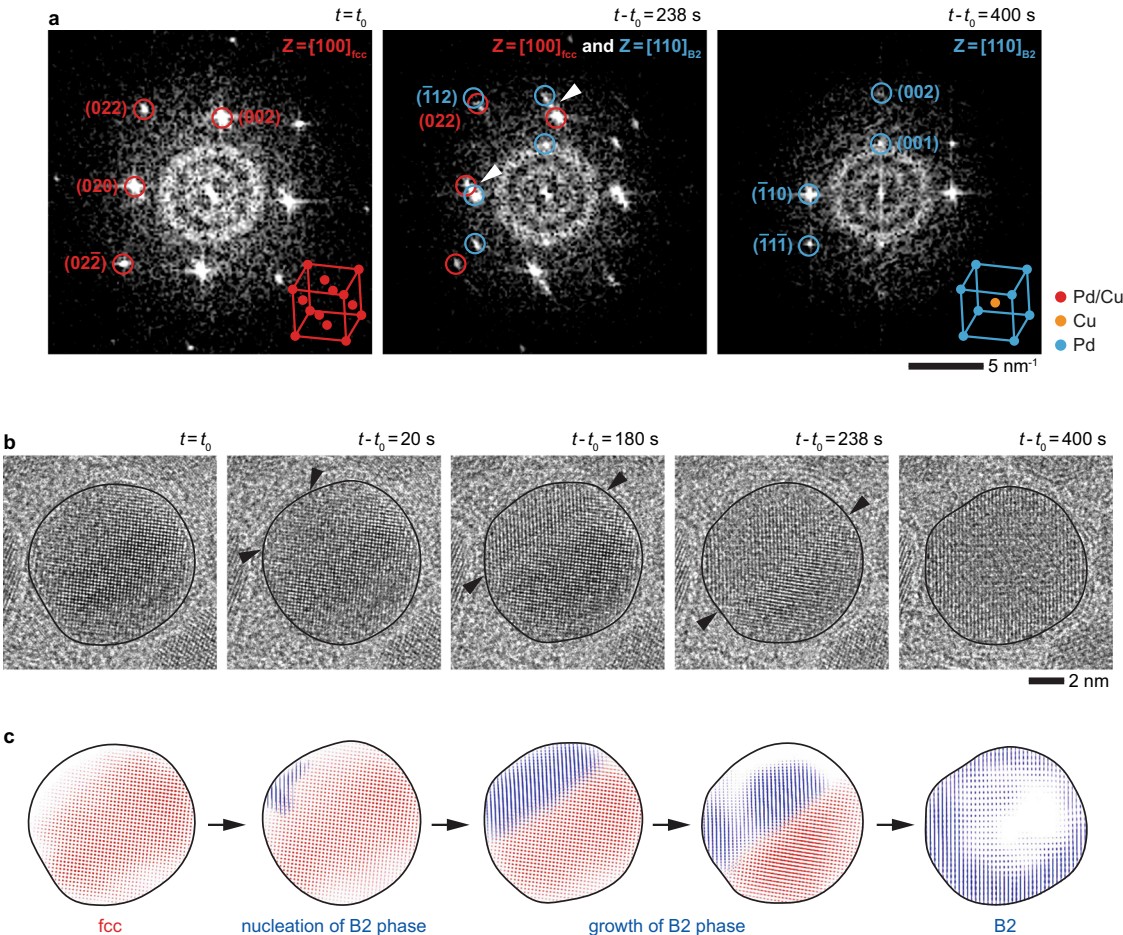

**Fig. 1 | Phase transition of an fcc PdCu alloy NP into a B2 NP. a** FFT and **b** TEM image series of a disordered fcc PdCu NP transforming into a B2 (i.e., ordered bcc) NP at 500 °C when viewed from $[100]_{fcc}$ and $[110]_{B2}$ zone axes (Supplementary Movie 1). The insets in **a** show schematics of fcc ($t = t_0$) and B2 ($t-t_0 = 400$ s) unit cells. The white arrows in **a** point to the traces connecting the fcc and B2 spots. The black arrows in **b** indicate the locations of the moving fcc−B2 interface during heating. **c** Sequence of inverse FFT images corresponding to the image series in **b** identifying the fcc (red) and B2 (blue) regions during the NP transformation. $t_0$ is the timepoint at which we started recording the process.

study. We chose this NP system for two reasons. First, the timescale for an fcc-to-bcc phase transition in PdCu alloy is sufficiently slow compared to steel[12,19], which, in turn, allows us to image the dynamics of the transition with current state-of-the-art cameras. Second, the stress from the surfaces of single-crystalline NPs is significantly lower than the stress on individual grains of polycrystalline metals constrained by the surrounding grains[32], enabling to probe the phase transition isolated from any external effects. A similar in situ TEM approach has been used to study many different phase transition phenomena in single-crystalline Pd[33], AgPd[34], Cu₂S[35], HfO₂[36] nanocrystals, and many other systems[37–40].

## Results

### Phase transition dynamics of an individual NP

We prepared PdCu alloy NPs with a metastable fcc phase using a one-pot synthesis reaction, in which Pd²⁺ and Cu²⁺ precursor ions were reduced simultaneously to form the alloy NPs[16]. The NPs comprised 46% (at.) Pd and 54% (at.) Cu and had an average size of ~8 nm (Supplementary Fig. 1). At this composition and below 505 °C, the bcc phase is a thermodynamically stable phase (Supplementary Fig. 2)[41]. Thus, upon sufficient heating, the as-synthesized metastable fcc NPs should transform into stable bcc NPs[16].

Figure 1 shows the structural transformation of an individual NP at 500 °C. The spots in the fast Fourier transform (FFT) pattern of the initial NP (Fig. 1a: $t = t_0$) correspond to a single-crystalline fcc phase. In

an fcc unit cell of PdCu alloy (inset in Fig. 1a: $t = t_0$), the Pd and Cu atoms randomly occupy the eight corners and six face-centered positions of the cell. After heating (to 500 °C), the fcc NP evolves into a bcc NP as seen from its FFT spots (Fig. 1a: $t-t_0 = 400$ s). Also, the presence of the forbidden $(001)_{bcc}$ spot suggests that the alloy NP has an ordered bcc crystal structure[42]. Specifically, in this bcc unit cell designated as the B2 phase (inset in Fig. 1a: $t-t_0 = 400$ s), the Cu atom occupies the body-centered position while the Pd atoms are at the eight corners of the unit cell[41]. During the fcc-to-B2 transition (Fig. 1a: $t -t_0 = 238$ s), the two different sets of spots in the FFT pattern correspond to coexisting fcc and B2 phases. Furthermore, the individual spots from two different phases are not distinctly separated; instead, they are connected with smeared traces (e.g., see traces between $(020)_{fcc} - (\bar{1}10)_{B2}$ and $(002)_{fcc} - (002)_{B2}$ spot pairs highlighted with white arrows in Fig. 1a: $t-t_0 = 238$ s). These traces between the spots of the two phases indicate that the fcc structure transforms into B2 structure gradually.

TEM and inverse FFT images of the NP in Fig. 1b, c show how the fcc phase evolves into the B2 phase. The B2 phase nucleates on the NP surface (Fig. 1b, c: $t-t_0 = 0-20$ s) and propagates across the NP (Fig. 1b, c: $t-t_0 = 20-400$ s) until it fully transforms into a B2 NP. Note that in this image series viewed from $[100]_{fcc}$ and $[110]_{B2}$ zone axes, the two phases are separated by a well-defined sharp interface. However, the interface is less clear when the NPs are viewed from other directions due to the overlapping grains of these two phases (Supplementary

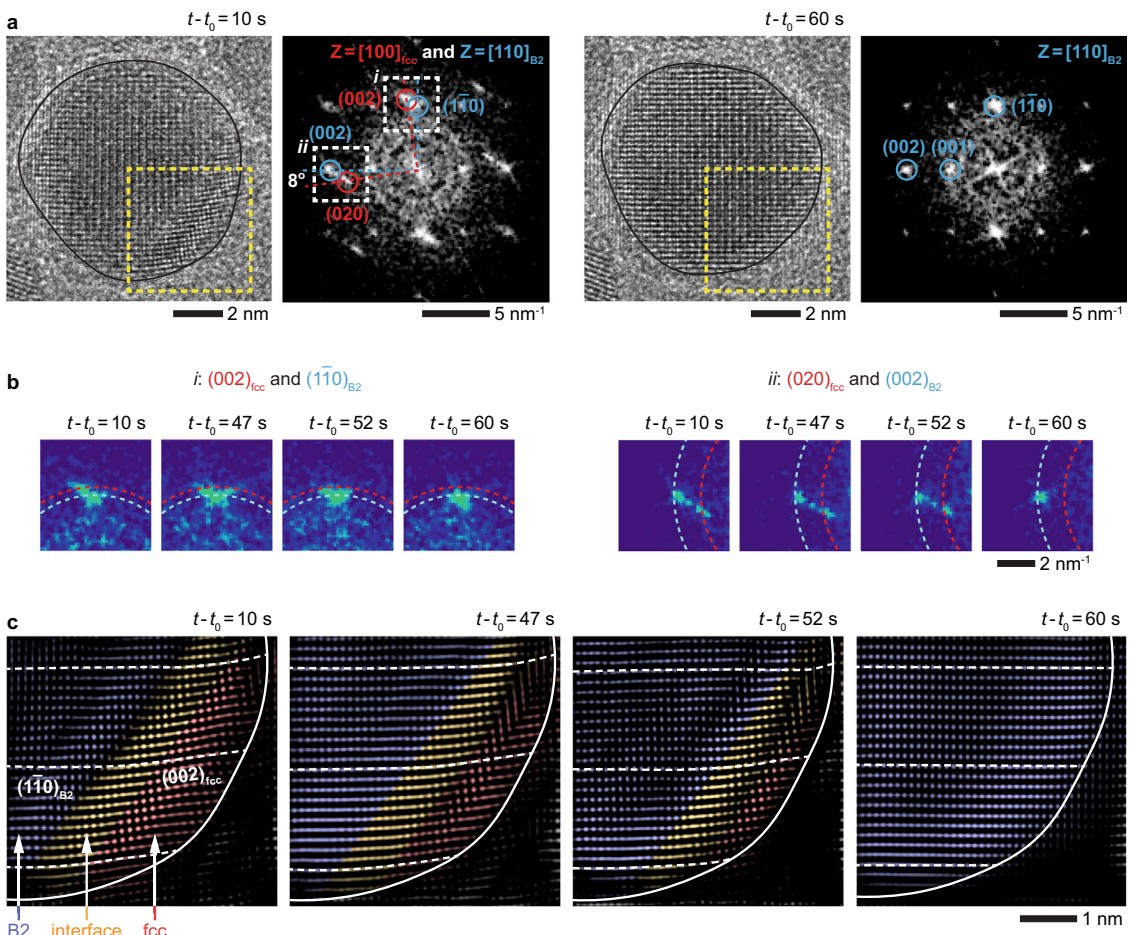

**Fig. 2 | Structure and dynamics of fcc−B2 interface. a** TEM and FFT images (from the dashed yellow boxes) showing the movement of the interface in the NP ($t$−$t_0$ = 10 s) as it converts into a B2 PdCu alloy NP ($t$−$t_0$ = 60 s) at 500 °C. **b** The two enlarged views marked (with dashed white boxes) in **a** showing the trace features connecting fcc and B2 spots (($002)_{fcc}$–$(1\bar{1}0)_{B2}$ spots and $(020)_{fcc}$–$(002)_{B2}$ spots) during the phase transition. The dashed red and blue arcs correspond to the reciprocals of fcc and B2 lattice spacings, respectively (i.e., $k_{(200)_{fcc}}$ = 5.3 nm$^{-1}$, $k_{(1\bar{1}0)_{B2}}$ = 4.8 nm$^{-1}$, and $k_{(200)_{B2}}$ = 6.7 nm$^{-1}$). **c** Sequence of inverse FFT images from the dashed yellow boxes in **a** showing the movement of the fcc−B2 interface (Supplementary Movie 4). (see Supplementary Fig. 7a for the unprocessed TEM images). The fcc, B2, and interface regions are false-colored in red, blue, and orange, respectively. The dashed white lines are guides showing how $(002)_{fcc}$ and $(1\bar{1}0)_{B2}$ planes connect via the interface. $t_0$ is the timepoint at which we started recording the process.

Figs. 3 and 4), implying that the fcc−B2 interface is nearly along the $(022)_{fcc}$ and $(\bar{1}12)_{B2}$ crystal planes.

### Structure of fcc−B2 interface and its movement

Figure 2a shows another example of a NP undergoing a similar fcc-to-B2 transition. A closer examination of the FFT (Fig. 2b) and inverse FFT (Fig. 2c) of the TEM image series reveals that smeared traces between the FFT spots of the two phases correspond to an ~9-Å-wide interface between these phases (Supplementary Note 3). This finite-width interface is a region within which the crystal planes evolve from $(002)_{fcc}$ ($d_{(002)_{fcc}}$ = 1.9 Å) into $(1\bar{1}0)_{B2}$ planes ($d_{(1\bar{1}0)_{B2}}$ = 2.1 Å), with the angle between these planes being roughly 8° (Fig. 2a–c: $t$−$t_0$ = 10 s).

The few-atomic-layers-wide interface between the fcc and B2 phases is a coherent interface, meaning the lattice planes of the two phases are continuous across the interface (Fig. 2c and Supplementary Fig. 11c). This is notable because a coherent interface forms only when the two different phases have the same crystal structure with less than 10% lattice mismatch[43,44]. Earlier studies describing the fcc-to-bcc phase transition in steels, where the lattice mismatch between the two lattices is large, propose that these transformations should proceed via a highly mobile semicoherent interface and a set of dislocations[17,45]. Contrary to this general expectation, the fcc−B2 interface in PdCu

system (with a large lattice mismatch of >20%) is coherent as it moves across the NP during the phase transition. The significance of the observed coherent interfaces is that the absence of defects between two phases is likely to be the reason as to why displacive transformations, common to steel[46] and shape-memory alloys[14], proceed at such high speeds[12].

### Discussion of the phase transition pathway

The B2 phase propagates into the fcc phase by deforming the fcc crystal, as seen from the movement of the interface in Fig. 2c. This continuous lattice deformation during the phase propagation can also be verified from other view directions (Supplementary Fig. 5). To obtain a three-dimensional perspective by unifying the observed structural changes viewed from different directions, we identified the fcc−B2 orientation relationship (see Supplementary Note 5 for more details) and summarized the phase transition pathway in Fig. 3. Here, a direct correspondence between the fcc and B2 crystal structures is established by constructing a body-centered tetragonal (bct) unit cell from two adjacent fcc unit cells (Fig. 3a and Supplementary Fig. 6a), known as Bain correspondence[46]. During the phase transition, the constructed bct unit cell evolves into a B2 unit cell through a slight rotation and a change in the cell dimensions (from 2.7 × 2.7 × 3.8 Å$^3$ to

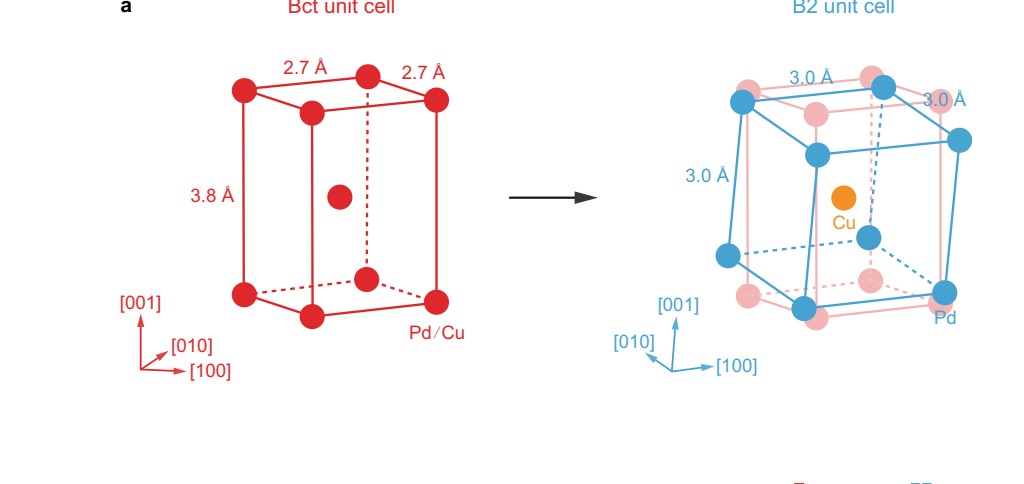

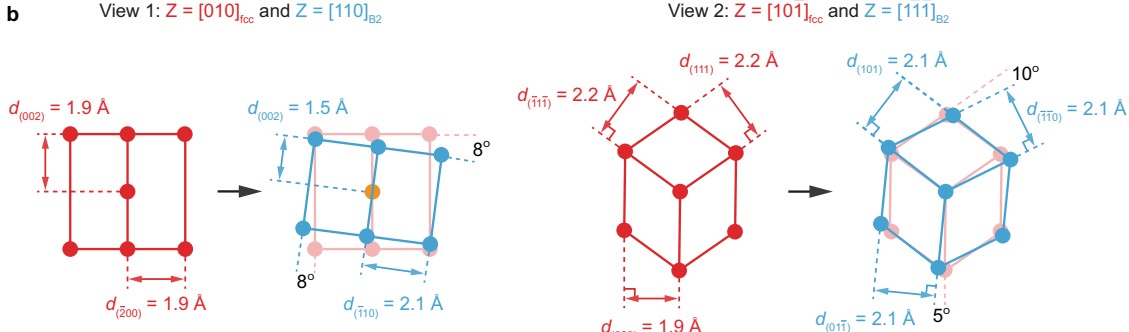

**Fig. 3 | Schematic of fcc-to-B2 phase transition. a** The atomic configurations of a body-centered tetragonal (bct) (red) and B2 (blue) unit cells with respective dimensions of $2.7 \times 2.7 \times 3.8$ Å$^3$ and $3.0 \times 3.0 \times 3.0$ Å$^3$. The bct unit cell is constructed from two disordered fcc unit cells with the dimension of $3.8 \times 3.8 \times 3.8$ Å$^3$ (Supplementary Fig. 6a). The fcc-to-B2 phase transition occurs via continuous rotation and change in the dimensions of the bct unit. **b** The projected views of the unit cells when viewed from two different directions, e.g., $[010]_{fcc}$ and $[110]_{B2}$ zone axes (View 1) and $[10\bar{1}]_{fcc}$ and $[1\bar{1}\bar{1}]_{B2}$ zone axes (View 2). Red spheres represent either Pd or Cu atoms in a disordered fcc unit cell, while blue and orange spheres correspond to Pd and Cu atoms in the B2 unit cell, respectively.

$3.0 \times 3.0 \times 3.0$ Å$^3$), producing the ordered final B2 unit cell as evidenced by the appearance of otherwise forbidden $(001)_{bcc}$ reflections (Fig. 1a). Here, Pd and Cu atoms undergo ordering into the B2 cell to occupy the corners and the body-centered position, respectively, via intra-cell atomic diffusion. Note that throughout the entire transformation, the cell volume does not change much (27.7 Å$^3$ vs. 27.0 Å$^3$; Fig. 3a).

The crystalline features of the fcc-to-B2 phase transition appear differently when viewed from different zone axes (Fig. 3b). When viewed from $[010]_{fcc}$ and $[110]_{B2}$ zone axes (Fig. 3b: View 1), $(002)_{fcc}$ planes change into $(002)_{B2}$ planes with the corresponding lattice spacings of 1.9 and 1.5 Å, and these lattice fringes rotate by ~8°. Also, $(\bar{2}00)_{fcc}$ planes become $(\bar{1}10)_{B2}$ planes with the respective lattice spacings of 1.9 and 2.1 Å. These features are in line with our TEM observations shown in Fig. 1, 2. Projection from $[10\bar{1}]_{fcc}$ and $[1\bar{1}\bar{1}]_{B2}$ zone axes (Fig. 3b: View 2) shows a 5° rotation between $(020)_{fcc}$ and $(01\bar{1})_{B2}$ planes, and a 10° rotation between $(111)_{fcc}$ and $(\bar{1}10)_{B2}$ planes (see Supplementary Note 2 for a more detailed discussion on how other view directions correlate with our TEM observations). It is important to note that Bain correspondence was proposed in 1924 and has been widely used ever since to describe the fcc−bcc phase transitions despite lacking direct experimental evidence for the transformation[45,46]. Our direct observations from different projected views reveal the details of Bain transformation.

To understand the fcc-to-B2 phase transition from a thermodynamic perspective, we evaluate the bulk and surface energies of fcc and B2 phases using density functional theory (DFT) calculations (Fig. 4 and Supplementary Table 2). The energy of the bulk B2 crystal is 36 meV/atom lower than that of the bulk fcc crystal (i.e.,

$E_{fcc \to B2} = E_{fcc} - E_{B2} = 36$ meV/atom), suggesting that the B2 phase is the stable phase. This is in line with previous studies[47,48] and the Cu−Pd phase diagram (Supplementary Note 1)[41]. To explain our observed phase transitions, we note that $E_{fcc \to B2}$ can be interpreted as the thermodynamic force driving the metastable fcc phase into B2 phase (at 350−500 °C). This driving force for the phase transition increases significantly in the presence of surfaces ($E_{\{hkl\}_{fcc} \to \{ijk\}_{B2}} = 64 - 304$ meV/atom) compared to the bulk case ($E_{fcc \to B2} = 36$ meV/atom), as shown in Fig. 4a, where commonly observed facets between the two phases are compared. Note that the transformation of $\{220\}_{fcc}$ into $\{211\}_{B2}$ surface has the highest driving force of 304 meV/atom. This is consistent with the TEM observation in Fig. 1, showing that the phase transition initiates and proceeds along the $(022)_{fcc}$ and $(\bar{1}12)_{B2}$ crystal planes. Furthermore, in line with general expectations, the driving force for the phase transition is larger for smaller NPs (with larger surface-to-volume ratios; Fig. 4b). In more general terms, the larger driving force for the surfaces implies that the new B2 phase should nucleate from the surface and not from the core of a NP.

## Nucleation of the B2 phase

In order to explore how an fcc-to-bcc phase transition initiates, we tracked the nucleation of a B2 phase within an fcc NP. During our heating studies, when the temperature was increased from 350 to 500 °C at a rate of 1 °C/s (Fig. 5a), the $(002)_{fcc}$ planes of the original fcc crystal (Fig. 5b: $t-t_0 = 12$ s) start to bend at the top left corner of the NP, as shown in Fig. 5b. First, the $(002)_{fcc}$ planes rotate by 2° (Fig. 5b: $t-t_0 = 40$ s), and then bend further until the angle reaches 5° (Fig. 5b: $t-t_0 = 73-167$ s). This new transient structure at the top-left corner of the NP is structurally identical to the coherent interface shown in

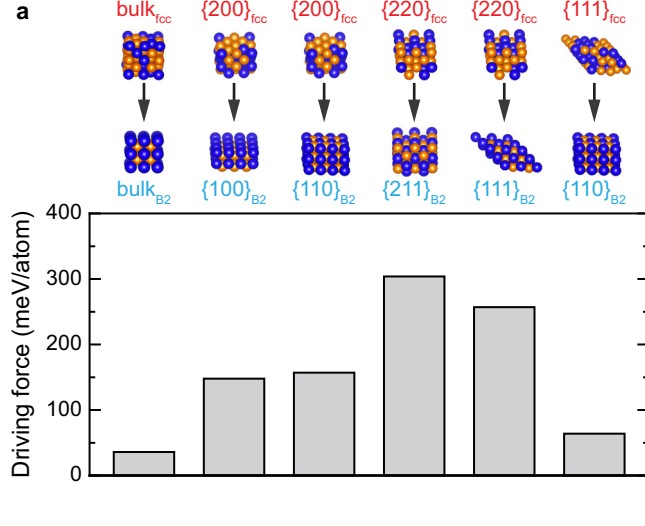

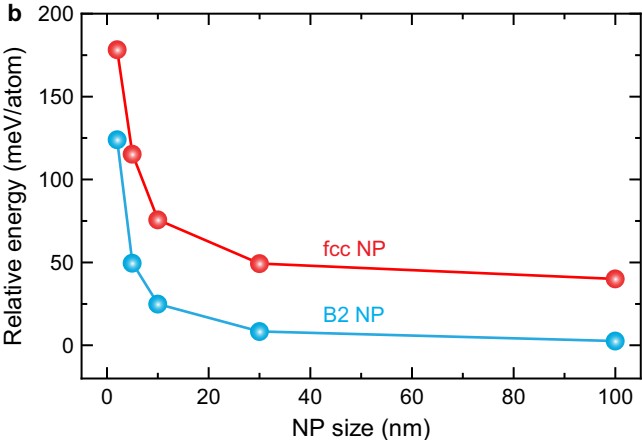

**Fig. 4 | Driving force for fcc-to-B2 phase transition. a** Driving forces or relative energies for the fcc-to-B2 phase transition in bulk crystal and for different surface planes. The relative energies were obtained using DFT calculations for the bulk supercells and surface slab models of fcc and B2 phases shown above the the plot. **b** Calculated relative energies of fcc and B2 NPs (with respect to the bulk B2 phase) as a function of NP size.

Fig. 2c and serves as a precursor phase which then transforms into a B2, and the growth direction of the B2 phase follows the propagation of the interface into the fcc NP. Note that the newly formed B2 phase clearly displays $(\bar{1}01)_{B2}$ and $(\bar{1}10)_{B2}$ lattice planes (Fig. 5a: $t-t_0 = 167$ s), and the transformation of the fcc into the B2 phase at the corner of the NP is consistent with the projected view expected from $[10\bar{1}]_{fcc}$ and $[1\bar{1}\bar{1}]_{B2}$ zone axes (Fig. 3b: View 2). The fact that coherent interface forms prior to the nucleation of the B2 phase, serving as a prenucleation phase for B2, is in stark contrast to earlier studies proposing the nucleation of a bcc phase to occur either via twining within the fcc structure[49], or the formation of stacking faults from fcc dislocations[50].

As a final remark, it is important to mention that single-crystalline NPs used in our study constitute the simplest model system, which while suitable to capture the essence of the phase transition, may deviate from the transitions that occur in bulk polycrystalline materials in several ways. In bulk, the nucleation usually starts from the grain boundaries, dislocations, or defects instead of free surfaces in the NP case. Second, the phase transitions and volume changes of the individual grains are constrained by the surrounding grains; thus, strains and dislocations could arise to accommodate the volume changes. Nevertheless, because of ultrahigh speeds at which many displacive

transformations take place, it is very likely that (dislocation-free) coherent interfaces are more common in these phase transitions. Furthermore, our study shows that such coherent interfaces not only play an important role in the propagation of a new phase during s−s phase transitions but are also critical for its nucleation as well.

## Methods

### Synthesis of fcc PdCu NPs

The following reagents were purchased from Sigma-Aldrich Co. (St Louis, MO, USA) unless otherwise noted: sodium tetrachloropalladate(II) ($Na_2PdCl_4$, 99.99%, Cat. No. 379808), copper(II) acetylacetonate ($Cu(C_5H_7O_2)_2$, 99.9%, Cat. No. 514365), D-(−)-ribose ($C_5H_{10}O_5$, 99%, Cat. No. R7500), 1-octadecene (ODE, $C_{18}H_{36}$, 90%, Cat. No. O806), oleylamine (OAm, $C_{18}H_{37}N$, 70%, Cat. No. O7805), hexane ($C_6H_{14}$, 99%, Cat. No. 32293), ethanol ($C_2H_6O$, 99.8%, Cat. No. 10437341, Fisher Scientific U.K. Ltd, Loughborough, UK).

The PdCu alloy NPs used in our study were synthesized based on a modified procedure of Tong et al.[16]. Specifically, 5.0 mg of $Na_2PdCl_4$, 6.6 mg of $Cu(acac)_2$, 11.0 mg of ribose, 1.3 mL of ODE, and 1.3 mL of OAm were mixed in a 20 mL capped glass vial and ultrasonicated for 30 min to dissolve the solid chemicals (i.e., $Na_2PdCl_4$, $Cu(acac)_2$ and ribose) in the mixed solvent solution (i.e., ODE and OAm). Next, the vial containing the homogeneous solution was heated from room temperature to 180 °C in an oil bath and kept at this temperature for 180 min during which the color of the final solution turned black. Then synthesized NPs were washed by centrifugation with hexane/ethanol mixture (at a volume ratio of 1:1) five times and then dispersed in hexane before the use.

### In situ TEM experiments

Two TEMs were used for in situ heating studies: Thermofisher Titan S/TEM (Thermo Fisher Scientific Ltd., Hillsboro, OR, USA) equipped with a Bruker Xflash 6 T | 30 energy-dispersive X-ray (EDX) spectrometer (Bruker, Billerica, MA, USA) and an aberration-corrected JEOL ARM300F (JEOL Ltd., Tokyo, Japan). Both TEMs were operated with an accelerating voltage of 300 kV. The Wildfire heating holders (DENS-solutions, Delft, Netherlands) were used for heating the NPs during in situ observations. Typical electron fluxes used for in situ imaging were in the range of 200–6000 e⁻ Å⁻² s⁻¹. Image series were acquired with two different cameras: a Gatan K2 IS camera (Gatan Inc., Pleasanton, CA, USA) on the Thermofisher Titan S/TEM and a Gatan One-View camera (Gatan Inc., Pleasanton, CA, USA) on the JEOL ARM300F. Inverse FFT images in Fig. 1c were obtained by separately filtering and then combining fcc and B2 spots in FFT patterns in Fig. 1a, and the same processing was used for Supplementary Figs. 3, 4, 10, and 14.

### DFT computations

All plane-wave DFT computations were performed with the PWSCF program available in the Quantum Espresso suite[51,52]. Here, Perdew−Burke−Ernzerhof (PBE) functional under the generalized gradient approximation (GGA) framework was used to approximate the exchange−correlation energies[53]. Electron−ion interactions were modeled with the projector augmented-wave (PAW) method. Pseudopotentials associated with the PBE functional and the PAW method were obtained for Cu and Pd from the pslibrary[54]. The kinetic energy cutoffs for wavefunctions and charge densities were set to 50 and 500 Ry, respectively. Gaussian smearings, necessary for modeling metallic systems, were employed with a width of 0.01 Ry. The bulk B2 phase was modeled with the experimentally reported 3.0 Å × 3.0 Å × 3.0 Å unit cell with 8 × 8 × 8 Monkhorst−Pack (MP) sampling of the Brillouin zone[55]. To mimic a disordered fcc structure, we modeled it with a 32-atom special quasi-random structure (SQS) supercell with 4 × 4 × 4 MP sampling[56,57]. The surface slabs for fcc and B2 structures were generated using the *pymatgen* package[58] with a minimum slab thickness of 5 Å and minimum vacuum thickness of 10 Å. The MP sampling of the

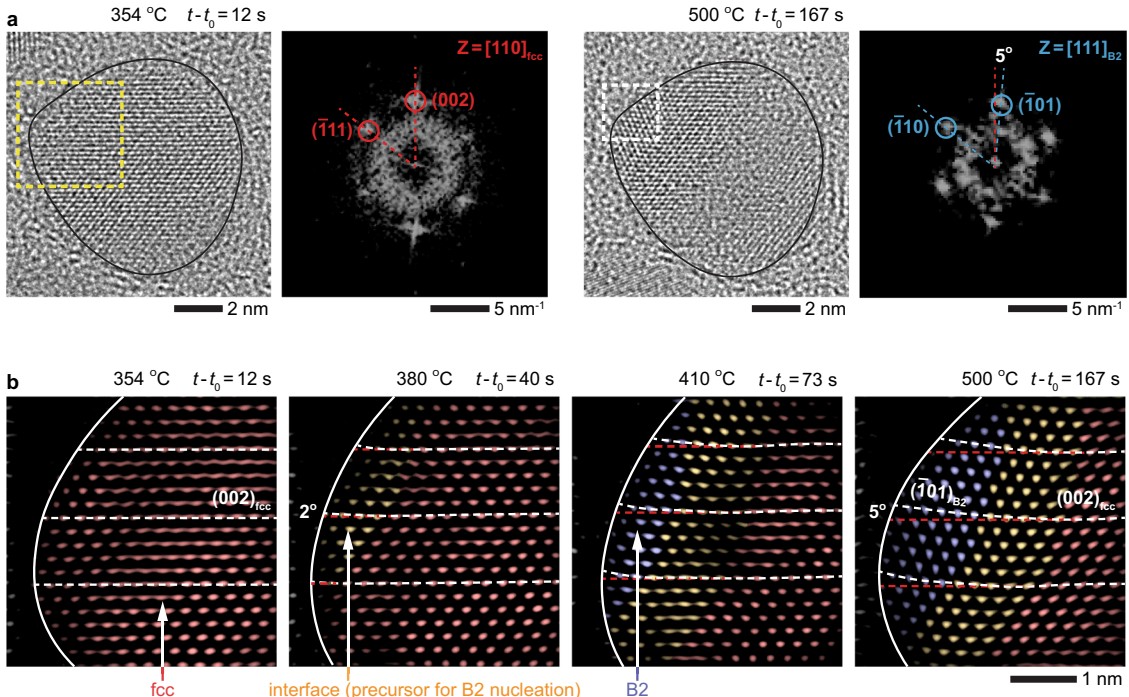

**Fig. 5 | Nucleation of B2 phase in an fcc crystal. a** TEM and FFT images (from the dashed yellow and white boxes) showing the nucleation of a B2 phase ($t-t_0 = 167$ s) on the top left corner surface of an fcc PdCu NP ($t-t_0 = 12$ s). **b** Sequence of inverse FFT images of the area (dashed yellow box) in **a** showing the nucleation ($t-t_0 = 12-73$ s) and growth ($t-t_0 = 167$ s) of the B2 phase (Supplementary Movie 6). (see Supplementary Fig. 9a for the unprocessed TEM images). The fcc, B2, and interface regions are false-colored in red, blue, and orange, respectively. Note

that the nucleation of the B2 phase starts with the formation of an interface region, which acts as a precursor phase for the nucleation. The dashed white lines are guides showing the gradual bending of $(002)_{fcc}$ planes into $(\bar{1}01)_{B2}$ planes as the B2 phase nucleates and grows. The dashed red lines are the $(002)_{fcc}$ planes. $t_0$ is the timepoint at which we started increasing the temperature from 350 to 500 °C at a rate of 1 °C s$^{-1}$.

respective surface slabs was determined based on the reciprocal of the ratio of supercell dimensions constructed with *pymatgen*. Following are the corresponding MP *k*-point meshes used for each of the slabs: $8 \times 8 \times 1$ for $\{100\}_{B2}$, $10 \times 10 \times 1$ for $\{110\}_{B2}$, $12 \times 12 \times 1$ for $\{111\}_{B2}$, $2 \times 2 \times 1$ for $\{211\}_{B2}$, $5 \times 5 \times 1$ for $\{200\}_{fcc}$, $6 \times 4 \times 1$ for $\{220\}_{fcc}$, and $5 \times 5 \times 1$ for $\{111\}_{fcc}$. To estimate the relative energies of individual NPs, we used the fcc NPs closed with $\{111\}$, $\{200\}$, and $\{220\}$ surfaces (with the surface ratios of $A_{\{111\}_{fcc}} : A_{\{200\}_{fcc}} : A_{\{220\}_{fcc}} = 1 : 1 : 1$ and the average surface energy of $\bar{E}_A = 399$ meV/atom) and the B2 NPs enclosed with $\{100\}$, $\{110\}$, and $\{211\}$ surfaces (with the surface ratios of $A_{\{100\}_{B2}} : A_{\{110\}_{B2}} : A_{\{211\}_{B2}} = 1 : 1 : 1$ and the average surface energy of $\bar{E}_A = 227$ meV/atom).

### Reporting summary
Further information on research design is available in the Nature Portfolio Reporting Summary linked to this article.

## Data availability
The data that support the findings of this study are available from the corresponding author upon request.

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

## Acknowledgements

This work was supported by the Singapore National Research Foundation's Competitive Research Program funding (NRF-CRP23-2019-0001). M.D. acknowledges the Ministry of Education of Singapore for support through MOE Tier 1 RG87/19. The DFT computations carried out in this study was supported by the A*STAR Computational Resource Centre through the use of its high performance computing facilities. We thank Dr. Cyril Cayron from Ecole Polytechnique Fédérale de Lausanne (EPFL) for fruitful discussions.

## Author contributions

Y.J. and U.M. conceived the research. Y.J. prepared the samples and performed the TEM studies with M.D. S.J.A. and T.L.T. performed the DFT calculations. H.Y. performed image processing of TEM results. Y.J. and U.M. prepared the manuscript with input from all the other authors.

## Competing interests

The authors declare no competing interests.
