## [Peer Review File · Nature Communications]

Reviewer comments, first round review –

Reviewer #1 (Remarks to the Author):

This manuscript reports the authors' experimental work on FCC to BCC transition in a Pd-Cu alloy using the advanced in-situ TEM technique. The discovery of an intermedium layer between the FCC phase and the BCC phase is claimed as the major novelty of the work. Overall, the experimental work has been very well done, demonstrating the authors' good skill in in-situ TEM. But, the outcomes seem not sufficient for publication in Nature Communications for the following reasons:

First, there are a few errors related to fundamental of materials science:

1. The first sentence of the Introduction, "A solid–solid (s–s) phase transition is the transformation of a solid from one crystal structure into another, during which its elemental composition remains unchanged", is wrong. Only martensitic transformation can retain the same composition of the product as the parent. Most solid-solid phase transformations are associated with change in composition, such as the common precipitation and pearlitic transformation in steels.
2. In lines 42-44, the statement, "In addition, because fcc-to-bcc phase transition in steels, one of the most commonly studied systems, occurs at ultrahigh speeds", is wrong. Only martensitic occurs at high speed. Pearlitic transformation is a typically diffusion controlled. Some bainitic transformation takes days.

In sufficient novelties:

3. As stated in lines 57 and 58, in-situ TEM has been used to study solid-solid phase transformations in many other alloy systems.
4. More importantly, study interfacial structure between two crystals in TEM normally needs edge-on condition to avoid the overlap of the two crystals. In this work, as the BCC phase nucleates on the surface of the FCC spherical nanoparticles, edge-on condition is hard to achieved. Hence, there is not any evidence verifying the so-called interface between the FCC and BCC not the overlap of these two phases because it was not at edge-on.

One minor writing confusion

5. In lines 64 and 65, it states that "At this composition and below 505 °C, bcc phase is a thermodynamically stable phase". But, in lines 70 and 71, the statement is "After heating to 500°C, the fcc NP evolves into a bcc NP as seen from its FFT spots". Shall BCC be thermodynamically stable at lower temperature or higher temperature. It may confuse readers.

Reviewer #2 (Remarks to the Author):

Title: Dynamics of Fcc-to-Bcc Phase Transition in PdCu Alloy Nanoparticles Manuscript ID: NCOMMS-22-07566 Journal: Nature Communications

In the manuscript, authors have reported some interesting results on nanoscopic investigation of phase change dynamics (fcc-to-bcc phase transition) in PdCu alloy nanoparticles by using in situ heating transmission electron microscopy. The reported work will be of significance to the future researchers of the field. In general, the manuscript is well written, and the language is fluent. The presentation and organization of the work are very nice. The methodology presented here is clear and detailed enough.

Overall after careful evaluation of this manuscript, I believe that this article merits publication.

However, before publishing, the following quarries may be clarified by the authors.

Comments:

1. In the abstract authors have written "These insights into the fcc-to-bcc phase transition pathway are important for understanding solid–solid phase transitions in general and can help to

tailor the functional properties of metals and their alloys.”

Please elaborate and write specifically which properties can be tailored and highlight some of them in the abstract to draw the attention of future researchers.

2. Some detailed discussions and quantitative experimental results (bending, stacking faults, dislocations) about the phase transition of the alloy may be provided, if possible, for better clarity.

3. The thermodynamic calculations may be carried out to explain the phase change dynamics.

4. Molecular dynamics simulation establishes a bridge between theory and experimental observations. Author may like to provide MD simulation to understand the mechanisms of transformation in alloy at an atomistic scale, if possible.

5. Authors wrote, “direct correspondence between fcc and B2 crystal structures is established by constructing a body-centered tetragonal (bct) unit cell...” The authors should elucidate whether the interface layer actually consists of BCT unit cells or not. Furthermore, the Bain type transformation path and the diffusive nature of the transformation as discussed by the authors are not exactly cohesive. Author may like to read/follow the following articles, if found interesting, viz., (i) <https://doi.org/10.1088/1468-6996/15/2/025002> & (ii) Phys. Status Solidi A 2010, 207, 1874–1879, DOI: 10.1002/pssa.200925341.

6. The bct unit cell evolves into a B2 unit cell through slight rotation and change in the cell dimensions (from $2.7 \text{ \AA} \times 2.7 \text{ \AA} \times 3.8 \text{ \AA}$ to $3.0 \text{ \AA} \times 3.0 \text{ \AA} \times 3.0 \text{ \AA}$) The authors should also denote the cell parameters of the initial Fcc cell in this context.

7. A discussion regarding the observed double spots in the FFT images (Fig. 2A) should be included.

Reviewers #3 and #4 (Remarks to the Author)

This paper described the in situ observation of the phase transition of synthesized disordered FCC PdCu NPs to ordered BCC PdCu NP at 500C. HRTEM images and videos were reported, showing a distinct phase transition interface between FCC and BCC phases and propagation of BCC into the FCC phase of PdCu. FFT analysis confirmed the presence of the FCC phase before heating, FCC/BCC during propagation, and BCC phase at the end of the transition. The full transition is observed by 500C. The discovery of a coherent interface during the propagation process was unexpected due to the large lattice mismatch of the fcc-bcc interface in the PdCu system. Multiple zone axes were checked for the phase transition, and the analysis aided by structural simulations and Bain correspondence were consistent with the observations reported. The manuscript is interesting, the determination of a reproducible mechanism of phase transformation from fcc to bcc in PdCu NPs is of general relevance to the wider solid state materials community, and this work is worthy of publication after the following comments have been addressed:

1. The authors measure the width of the two-phase interface during transformation. How does the fact that TEM images are 2D projections limit their ability to make this measurement? Is it possible to estimate error bars?

2. The most surprising finding is that of a coherent interface between structures with a 20% mismatch. Is this made possible only because the system is nanoparticulate? If so, how does this finding translate to a bulk transformation mechanism? In the conclusions the authors state that they would still expect to see a coherent interface in a bulk sample; what is the basis for this statement?

Reviewer #5 (Remarks to the Author):

Jiang et al. have submitted a report on the fcc-to-bcc phase transition in PdCu nanoparticles with an average diameter of $\sim 8\text{nm}$. The authors showed that bcc nucleation consistently occurs at the surface of their nanoparticle model and that a coherent interface accompanies this transition. They have noted that the existence of a coherent interface is exciting and unexpected, particularly for a system with a high degree of lattice mismatch between the phases of interest, which is the case of PdCu.

The studies performed are well done, likely reproducible from the described methods, and represent, to the best of my knowledge, the only studies looking at this type of fcc-to-bcc phase transition with in situ TEM at the single-particle level in PdCu. Other recent works have investigated solid-solid phase transitions in alloys that would be appropriate to cite: see ACS Nano 2022, 16, 2, 1781–1790. I believe the authors could add several additional references to their final publication on similar phase transitions in other systems to be more complete. There is a high likelihood that this work could be relevant for other areas of materials science and chemistry, e.g., steel as noted by the authors, and therefore appropriate for publication.

While the use of small single-crystalline nanoparticles is a sound model system, several of the conclusions drawn in the manuscript could afford additional experimental evidence and/or clarification. Principally, the observation of the coherent phase transition could be exclusive to small, unstrained systems like individual single-crystal nanoparticles and should be tested against nanoparticles with identical compositions but larger sizes (30+nm). Based on the synthetic procedure presented in this manuscript, this experiment could be readily achieved by controlling nanoparticle nucleation and growth conditions. Larger particles would have a significantly lower surface-to-volume ratio and different interparticle strain, so testing such a system could support the universality of this coherent interface or reveal it to be an emergent phenomenon in extremely small systems. As particles become larger, they may also demonstrate discrepancies in bcc nucleation sites, which is another essential clarification to make to support the conclusions of this work entirely.

Reviewer #1:

This manuscript reports the authors' experimental work on FCC to BCC transition in a Pd-Cu alloy using the advanced in-situ TEM technique. The discovery of an intermedium layer between the FCC phase and the BCC phase is claimed as the major novelty of the work. Overall, the experimental work has been very well done, demonstrating the authors' good skill in in-situ TEM. But, the outcomes seem not sufficient for publication in Nature Communications for the following reasons:

Our response: We thank the reviewer for his/her positive comments on the manuscript, and we address the comments below.

First, there are a few errors related to fundamental of materials science:

1. The first sentence of the Introduction, "A solid–solid (s–s) phase transition is the transformation of a solid from one crystal structure into another, during which its elemental composition remains unchanged", is wrong. Only martensitic transformation can retain the same composition of the product as the parent. Most solid-solid phase transformations are associated with change in composition, such as the common precipitation and pearlitic transformation in steels.

Our response: We revised the introduction "A solid–solid (s–s) phase transition is the transformation of a solid from one crystal structure into another." (highlighted in yellow in the 2nd paragraph of page 1 of the main text) to make the description accurate.

2. In lines 42-44, the statement, "In addition, because fcc-to-bcc phase transition in steels, one of the most commonly studied systems, occurs at ultrahigh speeds", is wrong. Only martensitic occurs at high speed. Pearlitic transformation is a typically diffusion controlled. Some bainitic transformation takes days.

Our response: The sentence has been revised as "More importantly, in the case of martensitic transformations, which is a critical process step in steel production (e.g., displacive fcc-to-bcc transformations),²⁴ the phase transition occurs at ultrahigh speeds.^{12,25,26}" (highlighted in yellow in 2nd paragraph of page 2 of the main text) to correct the statement.

Insufficient novelties:

3. As stated in lines 57 and 58, in-situ TEM has been used to study solid-solid phase transformations in many other alloy systems.

Our response: Just to clarify, we do not claim the in situ TEM method to be the novelty of the work; the novelty lies in the detailed steps of the fcc-to-bcc phase transition. Specifically, contrary to earlier predictions of the interface between the two phases being semicoherent (ref. 17 and 45), our results clearly demonstrate that the interface is coherent, and that explains why displacive fcc-to-bcc phase transitions may propagate high speeds. Furthermore, for the first time, our observations show how a new bcc phase nucleates within the fcc phase and identifies the coherent interface as a prenucleation precursor for the nucleation. None of these concepts which govern the phase transition have been shown before. Hence, in our opinion, our results meet the requirements of the novelty.

4. More importantly, study interfacial structure between two crystals in TEM normally needs edge-on condition to avoid the overlap of the two crystals. In this work, as the BCC phase nucleates on the surface of the FCC spherical nanoparticles, edge-on condition is hard to achieve. Hence, there is not any evidence verifying the so-called interface between the FCC and BCC not the overlap of these two phases because it was not at edge-on.

Our response: The identified interface structure is not due to the overlapping of the two phases. The reciprocal spacing of a continuous “trace” (associated with interface) is distinctly different from the spacings associated with fcc and bcc phases (Figure 2), and this trace links the spots of the two phases. From the point of view of the Fourier transform, the overlap of two phases would still produce only spots associated with these phases. The fact that we can clearly identify the interface from reciprocal space both for side (Figure 2) and top-down (Supplementary Figure 5) shows that interface structure is real and continuous.

One minor writing confusion

5. In lines 64 and 65, it states that “At this composition and below 505 °C, bcc phase is a thermodynamically stable phase”. But, in lines 70 and 71, the statement is “After heating to 500 °C, the fcc NP evolves into a bcc NP as seen from its FFT spots”. Shall BCC be thermodynamically stable at lower temperature or higher temperature. It may confuse readers.

Our response: The as-synthesized NPs are metastable fcc NPs and heating (<505 °C) is needed for atoms to diffuse and for the NP to transform into the thermodynamically stable bcc phase. We revised the sentence for clarity (highlighted in yellow, last sentence/4th paragraph/page 2): “Thus, upon sufficient heating, the as-synthesized metastable fcc NPs should transform into stable bcc NPs.¹⁶”

Reviewer #2:

In the manuscript, authors have reported some interesting results on nanoscopic investigation of phase change dynamics (fcc-to-bcc phase transition) in PdCu alloy nanoparticles by using in situ heating transmission electron microscopy. The reported work will be of significance to the future researchers of the field. In general, the manuscript is well written, and the language is fluent. The presentation and organization of the work are very nice. The methodology presented here is clear and detailed enough.

Overall after careful evaluation of this manuscript, I believe that this article merits publication. However, before publishing, the following quarries may be clarified by the authors.

Our response: Thank you for the positive comments, and we address the questions below.

Comments:

1. In the abstract authors have written “These insights into the fcc-to-bcc phase transition pathway are important for understanding solid–solid phase transitions in general and can help to tailor the functional properties of metals and their alloys.” Please elaborate and write specifically which properties can be tailored and highlight some of them in the abstract to draw the attention of future researchers.

Our response: The Abstract and 1st and 3rd sentences of the Introduction (highlighted in yellow) elaborates on the specific properties and their utility in specific applications: “The phase transitions between these structures play an important role in the production of steel and high entropy alloys and the functioning of catalysts and shape memory alloys.”; “Because the crystal structure of a solid determines its mechanical strength,^{7,8} optical property,⁹ and electrical¹⁰ and thermal¹¹ conductivities, s–s transition is an important process in materials technology.¹²”; “The fcc-to-bcc phase transition and vice versa are often used in the production of durable steels,^{12, 13} shape memory alloys,¹⁴ high entropy alloys,¹⁵ and catalytic materials.¹⁶”

2. Some detailed discussions and quantitative experimental results (bending, stacking faults, dislocations) about the phase transition of the alloy may be provided, if possible, for better clarity.

Our response: We added a new **Supplementary Information Section 4** to provide detailed discussion and quantitative results on interface structure, which quantifies bending associated with all lattice columns (**Supplementary Figures 7 and 9**) and also tracks the evolution of the interface region during the phase transition (**Supplementary Figure 8**). Our analysis reveals that there are no stacking faults or any other noticeable defects in the interface region.

3. The thermodynamic calculations may be carried out to explain the phase change dynamics.

Our response: We used DFT calculations to identify the driving force behind the observed phase transition (**Figure 4**). Our interpretation of the results is described in detail in the 1st paragraph /page 8 (highlighted in yellow).

4. Molecular dynamics simulation establishes a bridge between theory and experimental observations. Author may like to provide MD simulation to understand the mechanisms of transformation in alloy at an atomistic scale, if possible.

Our response: It is extremely hard to capture the phase transition dynamics with MD simulations. The reason is that the typical timescale of MD simulation using the modest classical MEAM potential is $< 1\mu\text{s}$, and the timescale using DFT-based ab initio potentials is $< 1\text{ ns}$. However, the phase transition from experimental observations lasts for over 100 s. Thus, it is not possible for the state-of-the-art MD to reproduce the phase transition at such a timescale.

5. Authors wrote, “direct correspondence between fcc and B2 crystal structures is established by constructing a body-centered tetragonal (bct) unit cell...” The authors should elucidate whether the interface layer actually consists of BCT unit cells or not. Furthermore, the Bain type transformation path and the diffusive nature of the transformation as discussed by the authors are not exactly cohesive. Author may like to read/follow the following articles, if found interesting, viz., (i) <https://doi.org/10.1088/1468-6996/15/2/025002> & (ii) Phys. Status Solidi A 2010, 207, 1874–1879, DOI: 10.1002/pssa.200925341.

Our response: To elucidate the bct unit cell at the fcc–B2 interface, we added the schematic of bct, interface, and B2 unit cells in **Supplementary Figure 11C**, along with the corresponding description in the caption (highlighted in yellow).

Bain correspondence (Bain type transformation path): We agree with the referee that the phase transition of PdCu alloy is not purely displacive transformation; while atomic ordering in the alloy is diffusive (**Figure 3**), as in the 1st paper mentioned by the referee, restructuring of that lattice is displacive (supplementary ref. 5) While Bain correspondence is often used to describe diffusionless processes (as in the 2nd paper mentioned by the referee), the displacive restructuring that takes place in our unit cell warrants using the Bain correspondence as has previously been used in supplementary refs 6, 7 and 8. We elaborated on this in **Supplementary Information Section 3** (highlighted by yellow).

6. The bct unit cell evolves into a B2 unit cell through slight rotation and change in the cell dimensions (from $2.7\text{ \AA} \times 2.7\text{ \AA} \times 3.8\text{ \AA}$ to $3.0\text{ \AA} \times 3.0\text{ \AA} \times 3.0\text{ \AA}$) The authors should also denote the cell parameters of the initial Fcc cell in this context.

Our response: Now, we include the cell parameters in the caption of **Figure 3** (highlighted in yellow).

7. A discussion regarding the observed double spots in the FFT images (Fig. 2A) should be included.

Our response: **Figure 2B** is the enlarged views of the double spots in **Figure 2A**, which is discussed in the 2nd paragraph/page 4 (highlighted in yellow).

Reviewers #3 and #4:

This paper described the in situ observation of the phase transition of synthesized disordered FCC PdCu NPs to ordered BCC PdCu NP at 500 °C. HRTEM images and videos were reported, showing a distinct phase transition interface between FCC and BCC phases and propagation of BCC into the FCC phase of PdCu. FFT analysis confirmed the presence of the FCC phase before heating, FCC/BCC during propagation, and BCC phase at the end of the transition. The full transition is observed by 500 °C. The discovery of a coherent interface during the propagation process was unexpected due to the large lattice mismatch of the fcc-bcc interface in the PdCu system. Multiple zone axes were checked for the phase transition, and the analysis aided by structural simulations and Bain correspondence were consistent with the observations reported. The manuscript is interesting, the determination of a reproducible mechanism of phase transformation from fcc to bcc in PdCu NPs is of general relevance to the wider solid state materials community, and this work is worthy of publication after the following comments have been addressed:

Our response: Thank you for all the positive comments. Below we addressed your comments.

1. The authors measure the width of the two-phase interface during transformation. How does the fact that TEM images are 2D projections limit their ability to make this measurement? Is it possible to estimate error bars?

Our response: As shown in Figure 2C, the regions of the fcc and B2 phases are distinctly separated by the interface; thus, the interface is almost parallel to the incident electron beam, and the measured width should be very close to its actual value.

Following the referee's suggestion, we tracked the interface width during heating-induced movement (Supplementary Figure 8). The estimated interface width is $8.6 \pm 0.4 \text{ \AA}$.

2. The most surprising finding is that of a coherent interface between structures with a 20% mismatch. Is this made possible only because the system is nanoparticulate? If so, how does this finding translate to a bulk transformation mechanism? In the conclusions the authors state that they would still expect to see a coherent interface in a bulk sample; what is the basis for this statement?

Our response: Even though, during the revision, we extended our study to larger (30-nm) PdCu alloy NPs and found that our findings are applicable there as well (new Supplementary Information Section 7), it is possible that the phenomenon may be limited to nanoscale materials as shown in our manuscript. Our expectations that our finding can extend to bulk systems is based on two premises: 1) Most bulk alloys are made of nanoscale grains, which can be treated as NPs, albeit constrained but NPs nevertheless. 2) The fact martensitic fcc-to-bcc transformations occur at ultra-high speeds is indicative that the interface between the phases is likely coherent because semicoherent interfaces (*i.e.*, any interfacial dislocations) would significantly slow down the phase transition process. This being said, our expectation (in the conclusion) on bulk phase transitions is speculative at this point and will require more work to establish.

Reviewer #5:

Jiang et al. have submitted a report on the fcc-to-bcc phase transition in PdCu nanoparticles with an average diameter of ~ 8 nm. The authors showed that bcc nucleation consistently occurs at the surface of their nanoparticle model and that a coherent interface accompanies this transition. They have noted that the existence of a coherent interface is exciting and unexpected, particularly for a system with a high degree of lattice mismatch between the phases of interest, which is the case of PdCu.

The studies performed are well done, likely reproducible from the described methods, and represent, to the best of my knowledge, the only studies looking at this type of fcc-to-bcc phase transition with in situ TEM at the single-particle level in PdCu. Other recent works have investigated solid-solid phase transitions in alloys that would be appropriate to cite: see ACS Nano 2022, 16, 2, 1781–1790. I believe the authors could add several additional references to their final publication on similar phase transitions in other systems to be more complete. There is a high likelihood that this work could be relevant for other areas of materials science and chemistry, e.g., steel as noted by the authors, and therefore appropriate for publication.

Our response: We appreciate the positive comments! We now include this (ref 34) and five other references (refs 37–41) on phase transitions.

While the use of small single-crystalline nanoparticles is a sound model system, several of the conclusions drawn in the manuscript could afford additional experimental evidence and/or clarification. Principally, the observation of the coherent phase transition could be exclusive to small, unstrained systems like individual single-crystal nanoparticles and should be tested against nanoparticles with identical compositions but larger sizes ($30+\text{nm}$). Based on the synthetic procedure presented in this manuscript, this experiment could be readily achieved by controlling nanoparticle nucleation and growth conditions. Larger particles would have a significantly lower surface-to-volume ratio and different interparticle strain, so testing such a system could support the universality of this coherent interface or reveal it to be an emergent phenomenon in extremely small systems. As particles become larger, they may also demonstrate discrepancies in bcc nucleation sites, which is another essential clarification to make to support the conclusions of this work entirely.

Our response: Following the reviewer's suggestion, we synthesized larger (~ 30 nm) PdCu alloy NPs (Supplementary Figure 13). While imaging larger NPs is a bit challenging due to their thickness, these NPs behave the same (as now shown in Supplementary Figure 14) as the original 8 nm NPs during the phase transition. To obtain an even more detailed image of the interface, we chose to analyze the 22-nm NP (as now displayed in Supplementary Figure 15) because larger ones are very hard to see through; here, the interface structure is coherent and is similar to the interface of smaller (~ 8 nm) NPs (Figure 2C and Supplementary Figure 11).

Reviewer comments, second round review –

Reviewer #1 (Remarks to the Author):

The authors have done excellent research work, clearly showing the atomic coincidence during FCC to BCC transition in solids. But, the novelty is still not convincing even though the fundamental errors are fixed in the revised version.

The major novelty claimed is the observed interface that can act as an intermediate precursor for nucleation of BCC on FCC. It is not convincing because of the following reasons:

1. As mentioned in the initial review, the interface could be an overlap boundary if it were not edge-on. The curvature of a nanoparticle is so big and makes it hard to avoid the overlap. Unless such interface can also be observed in a bulk phase transformation at edge-on condition, interface overlap is possible when transformation occurs on a nanoparticle.
2. Even though the possibility of interface overlap is ignored, the observed interface could be a result of transformation strain. Due to the difference in atom packing density between FCC and BCC, there is always a transformation strain across the interface. The excellence of this work is clearly to show such strain in TEM. But, it is hard to be convinced that such interface is a precursor of the phase transformation.
3. It is a common knowledge that displacive transformation is associated with full atomic coherency and strain. It is in the textbook. Hence, the statement in the last a few sentences on page 9 are not necessary. In addition, the atomic coherency is a result of displacive transformation rather than a critical condition of transformation as stated in the manuscript.
4. Bain model has been widely used to describe the FCC to BCC transition. It includes lattice change and lattice parameter adjustment. But, Bain transition has never been experimentally verified correct or incorrect. If this work clarified this unclearness, it would make breakthrough.
5. It is a bit confusing in the caption for Figure 2. It states that the dashed red and blue arcs correspond to the reciprocals of fcc and B2 lattice spacings, respectively. As we all know, plane (hkl) in real space is shown as a point or commonly relrod in reciprocal space. How can two arcs be used to represent plane parallelism?

Reviewer #2 (Remarks to the Author):

In the present work authors have reported some interesting results on investigation of fcc-to-bcc phase change dynamics in PdCu alloy nanoparticles by using in situ heating TEM. In the revised versions authors have responded to most of the queries raised by the referees and authors have revised the manuscript to a large extent. The revised manuscript may be accepted for publication, as per my understanding.

Reviewer #3 (Remarks to the Author):

In our opinion the authors have adequately addressed all of the reviewer comments and the manuscript is now ready for publication.

Reviewer #5 (Remarks to the Author):

The updated manuscript by Jiang et al. has addressed all of my original comments, and I believe is appropriate to publish as is in Nature Communications.

Reviewer #1:

The authors have done excellent research work, clearly showing the atomic coincidence during FCC to BCC transition in solids. But, the novelty is still not convincing even though the fundamental errors are fixed in the revised version.

The major novelty claimed is the observed interface that can act as an intermediate precursor for the nucleation of BCC on FCC. It is not convincing because of the following reasons:

Our response: We thank the reviewer for his/her positive comments, and we addressed the comments below.

1. As mentioned in the initial review, the interface could be an overlap boundary if it were not edge-on. The curvature of a nanoparticle is so big and makes it hard to avoid the overlap. Unless such interface can also be observed in a bulk phase transformation at edge-on condition, interface overlap is possible when transformation occurs on a nanoparticle.

Our response: To compare our results with an overlapped region of the two phases, we built such a model and simulated its TEM images (Figure R1). The changes in the lattices connecting the $(002)_{\text{fcc}}$ and $(1\bar{1}0)_{\text{B2}}$ planes are largely discontinuous, as indicated by the dashed white lines (Figure R1A), and this feature remains unchanged when imaged under different off-axis tilting conditions (Figure R1B). Furthermore, in the FFT image, the fcc and B2 spots are distinctly separated (Figure R1A). These two features of the overlapped region are essentially different from our results, as shown in Figures 1, 2, and 5. In contrast to this, the simulated TEM and corresponding FFT images of the interface model (Figure R2) are in perfect agreement with the experimentally observed interface between fcc and B2 phases (Figure 2, Supplementary Figures 11 and 15).

Figure R1. Simulated TEM images showing a region with overlapping fcc and B2 phases. (A) Atomic model of an overlapped fcc and B2 region and the corresponding simulated TEM and FFT images. The thickness of the model is 1 nm. (B) Simulated TEM images of the overlapped region when viewed from different off-axis tilting conditions. The dashed white lines are guides showing how (002)_{fcc} and (110)_{B2} planes connect. The white arrows in the FFT images shown in (A) highlight the disconnections of the fcc and B2 spots.

Figure R2. Simulated TEM images of an fcc–B2 interface as described in Figure 2. (A) Atomic model of an fcc–B2 interface and the corresponding simulated TEM and FFT images. The thickness of the model is 1 nm. (B) Simulated TEM images of the fcc–B2 interface when viewed from different off-axis tilting conditions. The dashed white lines are guides showing how $(002)_{\text{fcc}}$ and $(1\bar{1}0)_{\text{B2}}$ planes connect via the interface. The white arrows in the FFT images shown in (A) highlight the connections of the fcc and B2 spots.

2. Even though the possibility of interface overlap is ignored, the observed interface could be a result of transformation strain. Due to the difference in atom packing density between FCC and BCC, there is always a transformation strain across the interface. The excellence of this work is clearly to show such strain in TEM. But, it is hard to be convinced that such interface is a precursor of the phase transformation.

Our response: As shown in Figure 5B: $t - t_0 = 40$ s and Supplementary Figure 9C, the interface (precursor for B2 nucleation) formed prior to the emergence of the B2 phase when the $(002)_{\text{fcc}}$ planes bent only by $\approx 2^\circ$. Hence, the interface/precursor structure can exist without the B2 phase; thus, the interface is not the result of the strain between the two phases.

3. It is a common knowledge that displacive transformation is associated with full atomic coherency and strain. It is in the textbook. Hence, the statement in the last a few sentences on page 9 are not necessary. In addition, the atomic coherency is a result of displacive transformation rather than a critical condition of transformation as stated in the manuscript.

Our response: The reason for explicitly mentioning this is two-fold. The first reason is to remind the broader audience about the property of the interface. Second, only in the case of steel, it was *proposed* (and not observed) that the interface between austenite (fcc) and martensite (bcc) should be a glissile semicoherent interface with a set of parallel dislocations (ref. 17 and 45), and not in other systems. As for the displacive transformations in general, the high speed at which these transformations occur (ref. 12, 14, and 46) makes the direct experimental observations of the interface structures, interface movements, and nucleation dynamics extremely challenging; hence it has never been observed.

4. Bain model has been widely used to describe the FCC to BCC transition. It includes lattice change and lattice parameter adjustment. But, Bain transition has never been experimentally verified correct or incorrect. If this work clarified this unclarity, it would make breakthrough.

Our response: Exactly, and to our best knowledge, our results provide the first experimental observation that confirms the Bain model. To highlight this, *i.e.*, that our experimental results verified Bain model, we added the following sentence: “It is important to note that Bain correspondence has been proposed in 1924 and has been widely used ever since to describe the fcc–bcc phase transitions despite lacking direct experimental evidence for the transformation.^{45,46} Our direct observations from different projected views reveal the details of Bain transformation.” (highlighted in yellow in the 1st paragraph of page 7 of the main text).

5. It is a bit confusing in the caption for Figure 2. It states that the dashed red and blue arcs correspond to the reciprocals of fcc and B2 lattice spacings, respectively. As we all know, plane (hkl) in real space is shown as a point or commonly rod in reciprocal space. How can two arcs be used to represent plane parallelism?

Our response: The description has been revised as “The dashed red and blue arcs correspond to the reciprocals of fcc and B2 lattice spacings, respectively (*i.e.*, $k_{\{200\}_{\text{fcc}}} = 5.3 \text{ nm}^{-1}$, $k_{\{110\}_{\text{B2}}} = 4.8 \text{ nm}^{-1}$, and $k_{\{200\}_{\text{B2}}} = 6.7 \text{ nm}^{-1}$).” We used the arcs to highlight the existence of the lattice planes at different timepoints instead of their directions.

Reviewer comments, third round review –

Reviewer #1 (Remarks to the Author):

I am happy with the revision. Thanks